# Validation of the Mentalization Scale (MentS) in francophone control and clinical samples

Flora Descartes[1,2]*, Vincent Besch[1], Margaux Bouteloup[1,2], Rosetta Nicastro[3,4], Eléonore Pham[3,4], Eva Rüfenacht[3,4], Nader Ali Perroud[3,4], Martin Debbané[1,5]*

**1** Developmental Clinical Psychology Research Unit, Faculty of Psychology and Educational Sciences, University of Geneva, Geneva, Switzerland, **2** Université Marie et Louis Pasteur, UFR SLHS, Besançon, France, **3** Faculty of Medicine, University of Geneva, Geneva, Switzerland, **4** Research Department of Clinical, Educational and Health Psychology, University College London, London, United Kingdom, **5** Division of Psychiatric Specialties, Department of Psychiatry, Geneva University Hospitals, Geneva, Switzerland

* flora.descartes@unige.ch (FD); martin.debbane@unige.ch (MD)

## Abstract

### Aims

The Mentalization Scale (MentS) is a self-report measure for the assessment of mentalizing capacities, consisting of 28 items, yielding a three-factor structure: self-mentalizing, mentalizing others and motivation to mentalize. Its anglophone version has been validated for usage in both research and clinical contexts. The present study explores the psychometric properties of the francophone translation of the MentS, in both control and clinical samples.

### Method

A total of 711 participants were enrolled in this study. The MentS was administered to a community sample (N = 302, 161 females, Mage = 37.1, SDage = 12.3), and to a clinical sample composed of participants diagnosed with borderline personality disorder (BPD), attention deficit hyperactive disorder (ADHD) and co-occurring BPD and ADHD (N = 409, 266 females, Mage = 32.5, SDage = 11.9). Confirmatory factory analysis was used to analyze the fit of the model in our data, followed by reliability and validity analyses.

### Results

Results from confirmatory factor analysis (CFA) revealed a 27-item model to best fit the data for both control and clinical samples. In the control sample, good internal consistency was found for the total scale (α = 0.856, ω = 0.867) as well as for the three subscales MentS-Motivation (α = 0.789, ω = 0.801), MentS-Other (α = 0.792, ω = 0.798) and MentS-Self (α = 0.824, ω = 0.828). Similarly, good internal consistency was found in the clinical sample for the total scale (α = 0.871, ω = 0.879) and subscales

**Data availability statement:** All relevant data are within the manuscript and its Supporting Information files.

**Funding:** The PI (Martin Debbané) was funded by the Swiss National Science Foundation (Grant No. 100014_179033). The funders had

no role in study design and data collection, analysis, or interpretation.

**Competing interests:** The authors have declared that no competing interests exist.

(Motivation α = 0.770, ω = 0.783) (Others α = 0.842, ω = 0.847) (Self α = 0.808, ω = 0.815). The MentS demonstrated good temporal stability over a one-year interval, with excellent average-measures ICC for the total scale (ICC = .877, 95% CI [.843,.904], $p < .001$), and strong reliability for the Motivation (.837), Others (.806), and Self (.837) subscales (all $ps < .001$). The validity of the scale was confirmed using additional measures showing a coherent pattern of associations in relation to components underlying the construct of mentalization: reflective functioning, childhood trauma, cognitive emotion regulation, overall psychopathology distress and borderline symptomatology.

## Conclusion & Clinical Implications

The MentS can be used for research and clinical purposes in francophone samples. Our results suggest that amongst French speaking samples, a 27-item solution may be most optimal in both clinical and control populations. Evidence shows the scale could be employed across diagnostic entities, and that participants scoring low in self-mentalizing (MentS-Self subscale) may be more likely to report increased manifestations of psychopathology.

## Introduction

Mentalization, or reflective functioning, is a capacity based on imaginative mental activity, defined as the awareness an individual has of mental states (feelings, thoughts, beliefs and wishes in oneself and others) underlying human behavior [1]. Mentalization supports self-regulation by fostering integrated and updated mental models of the affective and cognitive dimensions of human experience; these mental models critically assist the navigation of interpersonal and social interactions [2]. The centrality of mentalizing in human functioning is found to play a role in mental health and illness, which positions mentalizing as a key transdiagnostic psychological process [3,4]. Indeed, impairments in mentalizing have been linked to a range of mental health disorders, including borderline personality disorder (BPD) [5], attention deficit hyperactive disorder (ADHD) [6], psychosis-spectrum disorders [7] and post-traumatic stress disorder (PTSD) [8]. Mentalizing is also targeted by clinical psychotherapeutic interventions, namely within the framework of mentalization based treatments (MBT), developed for personality disorders, such as BPD [9], antisocial personality disorder (ASPD) [10] or narcissistic personality disorder (NPD) [11], as well as other disorders such as eating disorders [12], drug addiction comorbidity [13,14], psychotic spectrum disorders [15,16] or complex trauma [17,18]. Such clinical evidence accentuates the importance of developing fine-grained understanding of the different components of mentalizing, to better understand the specific mechanisms sustaining improvements in mental health.

In this vein, the evaluation of mentalization has evolved significantly since the initial assessment of mentalizing based on the Adult Attachment Interview (AAI)

[19–21], which employed an hour-long semi-structured interview to explore attachment-related autobiographical memories. In this context, the Reflective Functioning Scale (RFS) was applied to provide specific scores on the basis of the participant's AAI transcript [22]. While highly informative and widely used as a gold-standard tool for assessing reflective functioning, the RFS is time-intensive, resource-heavy, and requires specialized training for clinicians, limiting its feasibility for large-scale research and daily clinical applications [23–25]. Additionally, although the RFS assesses multiple qualitative aspects of reflective functioning such as plausibility, complexity, and consistency, its scoring system remains unidimensional, thus limiting the interpretability of scores, and preventing factorial statistical examination of its underlying structure [23].

A significant development in the assessment of mentalization was introduced through the conceptualization of the dimensions of mentalizing, operationalized on the basis of neurofunctional dynamic systems involved in thinking about mental states [26]. Specific dimensions of mentalizing and their articulation reflects the involvement of distinct but coordinated systems: cognitive/affective mentalizing, self/other mentalizing, automatic/controlled mentalizing, internal/external mentalizing. This conceptual development fueled the creation of self-report measures to evaluate mentalizing. One of the first and most widely used self-report is the Reflective Functioning Questionnaire (RFQ), developed to assess the way in which an individual employs mental state information when mentalizing [27]. Among its currently validated versions (in at least 7 languages), the 8-item RFQ (RFQ-8) is thought to reflect both increased (certainty) and decreased (uncertainty) use of mental state information when mentalizing. The francophone validation of the RFQ-8 confirmed its two-factor structure and psychometric robustness, demonstrating relevance in studies of typical and atypical development and links to clinical phenomena such as non-suicidal self-injury [28]. In addition to the RFQ, other self-report instruments have been developed to assess distinct facets or domains of mentalizing. The Mentalization Questionnaire (MQ) explores facets of inefficient mentalizing (such as refusing self-reflection and psychic equivalence mode) as well as general factors such as emotional awareness and regulation of affect through mentalizing [29]. The Mentalized Affectivity Scale (MAS) examines the identification, processing and expression of affectively-laden mental states [30]. These instruments all attempt to operationalize mentalization that could address some of the previously observed limitations of the RFS. More recently, other self-report tools have also emerged and include the Cognitive and Affective Mentalization Scale Questionnaire (CAMSQ) [31], the Multidimensional Mentalizing Questionnaire (MMQ) [32], the Mentalizing Emotions Questionnaire [33], the Failure to Mentalize Trauma Questionnaire [34], and the Interactive Mentalizing Questionnaire [35].

In the context of self-report assessments of mentalization, although several tools have been developed, few instruments comprehensively fulfill the combined requirements of being time-efficient, easy to administer, reliable and dimension-specific. Furthermore, few have been validated across both community and clinical samples. The Mentalization Scale (MentS) is one of the only self-report instruments fulfilling these requirements [36]. The rationale behind the development of the MentS items aligns with what the authors have identified as key priorities in both research and clinical applications of mentalization. The items capture indicators of well-developed mentalizing capacity (e.g., explicit efforts to identify mental states or awareness of their subjective nature) [37], as well as markers of impaired or distorted mentalization (e.g., lack of interest in or imagination about mental states) [38,39]. Furthermore, the scale also considers global mentalizing, reflecting its developmental roots, connection to attachment quality, and relevance to overall mental health [38,40]. Consistent with the multidimensional nature of mentalization described above, factor analyses revealed three distinct dimensions: self-mentalizating, mentalizating others, and the motivational component underlying the drive to mentalize. The Motivation subscale (MentS-M; 10 items) assesses interest in understanding mental states, one's own and others', including curiosity and willingness to reflect (e.g., "I find it important to understand reasons for my behavior"). The Others subscale (MentS-O; 10 items) measures the ability to recognize and interpret others' emotions and intentions (e.g., "I can recognize other people's feelings"). The Self subscale (MentS-S; 8 items) captures the ability to access and reflect on one's own thoughts and feelings, including avoidance (e.g., "I do not like to think of my problems").

The MentS has been validated across several languages and populations, though the sample sizes and clinical groups in these studies vary widely. Among non-clinical populations, the MentS scale has been widely validated across various languages, including studies conducted in Iran [41,42], Poland [43], Italy [44], Korea [45], Turkey [46] and Japan [47], attesting to the robustness of the scale. All these studies consistently confirmed the three-factor structure of the scale, though some reported differences in factor loadings on specific items [45–47]. As far as studies with clinical samples are concerned, only two studies aside from the MentS princeps study [36] have included clinical samples. The Austrian validation assessed 26 psychiatric inpatients with various diagnoses [48]; their small sample size limits the validity and generalizability of the findings, underscoring the need for larger clinical samples. Similarly, the Chinese validation of the MentS in a sample of patients with schizophrenia (n = 200), demonstrated good internal consistency and reliability [49]. However, the focus on a single diagnostic group highlights the need for more diverse clinical populations and broader clinical validation.

The francophone validation of the MentS appears as a particularly relevant choice for the present study, given its multidimensional structure, existing cross-cultural validation, and focus on both self- and other-oriented mentalizing as well as the motivation towards mentalization. A further strength of the present study lies in the validation of the MentS within a large and diverse clinical sample. Most notably, individuals with borderline personality disorder (BPD) are included, thereby addressing a critical need for confirmatory evidence in this clinical population surveyed only in the original validation of the scale. Moreover, the sample encompassed individuals with other diagnostic profiles, namely ADHD and comorbid BPD and ADHD, which arguably augments the scale's potential for clinical application. The focus on a francophone population further contributes to the growing body of cross-linguistic validation efforts and underscores the cultural adaptability of the instrument.

Building on existing validation studies, the present study selected specific constructs were to assess the validity of the scale and its dimensions [36]. First, the total mentalizing score (MentS-Tot) reflects an individual's overall capacity for reflective functioning. Lower scores have been associated with both temporary (e.g., under emotional stress) and more enduring impairments in mentalization. These associations are well documented in borderline personality disorder (BPD) [3,50], and also reported in other conditions such as attention-deficit hyperactivity (ADHD) disorder and comorbid ADHD-BPD [6]. Given this, the MentS-Total score is expected to correlate negatively with borderline symptoms and overall psychopathology, as lower mentalizing capacity is a hallmark of various psychiatric conditions, particularly those involving emotion dysregulation and interpersonal dysfunction [3,36].

Second, the motivation to mentalize (MentS-M) has been associated with emotion regulation capacity [2,51]. Given that previous research has documented how the willingness to engage with one's own and others' mental states supports more flexible and constructive emotional processing and regulation, the MentS-Motivation subscale's score is expected to positively correlate with adaptive emotion regulation strategies, in line with other reports [2,37].

Third, mentalizing others (MentS-O) has been linked to RFQ-8 scores [44] and more specifically with significant correlations with certainty subscale [41]. This relation underlines a balanced usage of mental state information for social cognitive functioning [3]. This balance, neither overconfident nor excessively uncertain, is thought to support effective and adaptive mentalizing of self and others [1,28], and a recent longitudinal study reports its developmental association to prosociality and mental health [52]. Therefore, the MentS-Others score is expected to show positive associations with the certainty subscale of the Reflective Functioning Questionnaire, in line with prior work linking accurate social mentalizing with a confident, though not overly rigid, grasp of others' mental states [27,30].

Finally, self-mentalizing (MentS-S) has gained traction in recent mentalization research, particularly in understanding trauma-related disturbances [53–55]. Findings suggest that difficulties in self-mentalizing could be particularly evident in clinical populations [36,56]. For instance, individuals with a history of childhood trauma often present with fragmented internal representations, impairing their ability to make sense of their own experiences [57,58]. Impaired self-mentalizing can hinder coherent self-understanding and may lead to maladaptive self-attributions,

such as excessive self-blame [59]. As a result, lower MentS-Self scores are anticipated to negatively correlate with self-reported childhood trauma.

Based on these considerations, we hypothesized the original three factor 28-item model to fit the data well with acceptable to good fit indices and good internal consistencies for both the total and subscales in both clinical and control samples. In terms of reliability, we expected the scores for controls subjects to exhibit good temporal stability. For validity analyses, we predicted that the MentS-Tot scores would correlate negatively with general psychopathology expression in the control sample, and further with borderline symptom severity in the clinical sample. We predicted that the MentS-M scores would correlate positively with adaptive emotion regulation scores. We further expected that the MentS-O scores would correlate positively with the certainty subscale of reflective functioning. Finally, we hypothesized that the MentS–S scores would correlate negatively with self-reported childhood trauma.

## Method

### Participants

A total of 711 participants were enrolled in this study. First, a control sample with N = 302 adult participants (161 females, Mage = 37.1, SDage = 12.3, age range from 19–75). Participants, whose data were fully anonymized, were recruited on the Prolific website (https://www.prolific.com) during the month of March 2022 and they provided informed written consent. Inclusion criteria were to be older than 18, to be fluent in French, and to have never been hospitalized for psychiatric reasons. To ensure the quality of the data they provided, several inclusion and exclusion controls were applied. Participants were required to have satisfactorily submitted at least 15 online surveys beforehand to ensure a good level of use of the online platform and reduce data entry errors. Nonsensical items were randomly inserted in the survey and response times were monitored to check for participants' understanding and attention [60]. Participants were excluded if they provided incorrect responses to more than two bogus items or if their total response time was faster than the sample mean by more than two standard deviations (i.e., > 2 SD below the mean). In addition, participants who gave only one incorrect response on bogus items were also excluded if their total response time was at least one standard deviation below the sample mean. As a result of the combination of these rules, N = 70 participants were excluded.

Participants from the control group were contacted one year after the first survey; 257 replied and one was excluded for failing attention control, resulting in a retest subsample of n = 256 (84,77% of the initial sample) with 133 females and a mean age of 38.9 (SDage = 12.7) and who did not differ in age (Mann Whitney U = 5149, p = 0.175) and gender (c$^2$ [1] = 1.93, p = 0.165) in comparison to the baseline group.

Second, a clinical sample was composed of N = 409 participants (266 females, Mage = 32.5, SDage = 11.9, age range from 16−77). They were recruited at the emotional regulation disorder unit (ERD) at the University Hospitals of Geneva, which is a second and third line service specialized in the assessment and treatment of adult attention deficit hyperactivity disorder (ADHD) and borderline personality disorder (BPD) through evidence-based programs. Individuals in the clinical sample were diagnosed with three different principal diagnostics. The inclusion criteria for participation in the present study were being at least 18 years old, having a diagnosis of ADHD or BPD or co-occurring ADHD and BPD, and providing informed consent for participation in the study and use of health data for research purposes. Some were diagnosed with BPD (N = 133) as principal diagnosis, others presented with ADHD (N = 207) as principal diagnosis, and a portion of them had co-occurring diagnostics of BPD and ADHD (N = 69). Participants were assigned a battery of tests at arrival in the unit, including the MentS self-report questionnaire. Patient data was collected between 01/10/2020 and 29/01/2021 and accessed on the 20/02/2022 for research purposes. Authors had access to information that could identify individual participants during and after data collection. Informed and written consent was obtained by all participants at the time of admission in the unit. According to the Swiss law, parental agreement for participation of minors above 14 years of age is

not necessary. The study was approved by the Ethics Committee of the Geneva University Hospitals (no. 2021−00694) and by the Swiss Ethics Commission in Geneva under project BASEC id 2021−01100.

## Measures

*The Mentalization Scale (MentS; Dimitrijević, Hanak [36])*; is a self-report questionnaire designed to assess the capacity to mentalize. The original measure contains 28 items, which participants rate on a 5-point Likert scale (1 = completely untrue; 5 = completely true). As mentioned above the instruments yields 3 subscales reflecting domain-specific scores (MentS-Self, MentS-Others, MentS-Motivation), and the total score reflects the global mentalization capacity, with higher scores indicating stronger reflective functioning. The scale measures various aspects of mentalization through three subscales, namely the Motivation to mentalize, mentalizing Others and Self-mentalizing. The MentS anglophone version demonstrated respectively good and acceptable internal consistency with total-scale coefficients of α = 0.84 in control samples and α = 0.75 in clinical samples and subscales showed acceptable reliability (ranging from α = 0.74 to 0.79), except for the Motivation-subscale below the acceptable threshold (α = 0.60). The original English MentS was translated to French by independent French and English native speakers with the use of the forward-backward-forward procedure [61]. The French version of the scale can be found in the Supporting Information file S1 Data.

*The Symptom Checklist-90-Revised (SCL-90-R; [62]):* is a 90-item, self-report questionnaire that utilizes a 5-point Likert scale. It assesses nine primary symptom dimensions: Somatization, Obsessive-Compulsive, Interpersonal sensitivity, Depression, Anxiety, Hostility, Phobic Anxiety, Paranoid Ideation and Psychoticism. Additionally, it provides three scores reflecting global distress: Global Severity Index (GSI), Positive Symptom Distress Index and Positive Symptom Total. This scale has been validated among French-speaking populations, and was employed with the control group in this study. The scale has demonstrated stability and reliability for its main factors, including Depression, Somatization, and Panic-Agoraphobia, with Cronbach's alphas ranging from 0.77 to 0.90 across different subscales.

*The Borderline Symptom List (BSL-23; [63])*: is a short, 23-item self-rating assessment designed to measure borderline personality disorder typical symptomatology, employed in the clinical group in this study. The French version of the BSL-23 has demonstrated good psychometric properties, indicating its validity for both clinical and research purposes as it shows excellent internal consistency with Cronbach's α of 0.94 for the total score [64].

*The Cognitive Emotion Regulation Questionnaire (CERQ; [65]):* is a 36-item self-report, measuring nine different cognitive emotion regulation strategies, classified maladaptive and adaptive, that individuals use after experiencing negative events or situations. The CERQ focuses on an individual's thoughts rather than their actions. It is widely used for both research and diagnostic purposes. In this study, we focus on adaptive strategies (CERQ_Ad_ER), regarded as acceptance, positive refocusing, refocus on planning, positive reappraisal, and putting into perspective. The French version of the CERQ has demonstrated good psychometric properties, making it suitable for both research and clinical purposes with Cronbach's alphas ranging from 0.68 to 0.87 for the five adaptive strategies subscales [66].

*The 8-item Reflective Functioning Questionnaire (RFQ-8; [27]):* is a self-report, 8-item questionnaire. It consists of two subscales: certainty (RFQ_c) and uncertainty (RFQ_u) regarding mental states, with extreme scores on either subscale indicating impairments in mentalization. The French version of the questionnaire was validated in a non-clinical sample with α coefficients ranging from 0.70 to 0.83 for different subscales [28]. In this study, we employ a single dimension approach [67] based on the certainty subscale (RFQ_c) scoring scheme [52].

*The Childhood Trauma Questionnaire – Short Form (CTQ-SF; [68]):* is a 28 item, 5-point Likert scale, brief screening instrument designed to assess histories of maltreatment. Through five subscales scores, it evaluates the presence and severity of physical, emotional, and sexual abuse, as well as emotional and physical neglect. The French version of the CTQ-SF has been validated with good psychometric properties suitable for both clinical and research purposes with alphas ranging from 0.77 to 0.95 for the different subscales [69].

## Statistical analyses

**Descriptive statistics and data analyses.** Statistical analyses were conducted using Jamovi (desktop version 2.3.21.0) and IBM SPSS Statistics softwares (version 29.0.2.0 [20]). The characteristics of participants from both clinical and control samples were collected, including their gender, age, and diagnostic condition.

All variables were screened for missing data. A pragmatic, researcher-defined criterion was applied, consistent with prior research practices [70]: participants were excluded if more than 30% of items were missing on a given scale, and mean substitution was used when missingness was below this threshold. Twenty-five participants (N = 25) were excluded from analyses due to missing responses on more than 30% of items on one or more scales in the clinical sample and there were no missing values in the control sample.

Visual inspection of histograms, and boxplots was also performed to assess the shape of distributions and detect potential outliers. Extreme values were identified in the control group but not in the clinical sample. Specifically, one extreme low value was observed on the self-mentalizing subscale (MentS-Self), while no outliers were detected on the MentS-Others or MentS-Motivation subscales. On the MentS total scale, six extreme values were found: three on the lower end and three on the higher end. All data points, including these extreme cases, were retained in the analyses, as they were considered to reflect meaningful individual variation rather than error. As a result, nonparametric Spearman's rho correlations were computed to examine the associations among the MentS subscales in the control group to assess potential multicollinearity. A moderate positive correlation was found between MentS-Others and MentS-Motivation ($\rho$ = .703, p < .001), indicating substantial overlap between these two dimensions. Weaker but significant correlations were also observed between MentS-Self and MentS-Others ($\rho$ = .175, p = .002), and between MentS-Self and MentS-Motivation ($\rho$ = .120, p = .037). These results suggest that while the three subscales are related, they capture distinct facets of mentalizing. All correlations fell within acceptable ranges [71], indicating no significant concerns regarding multicollinearity.

**Confirmatory factor analysis.** Confirmatory factor analyses (CFA) were conducted using the SEMLj module as an interface to lavaan R package [72] in Jamovi (desktop version 2.3.21.0), with the robust Weighted Least Squares Mean and Variance adjusted (WLSMV) estimation method. The WLSMV estimator is well suited for ordinal data and provides robust estimates in the presence of non-normality and potential heteroscedasticity. CFA was first applied to the control sample to evaluate the proposed three-factor, 28-item model. Guided by the theoretical framework of mentalization and by modification indices, alternative models were tested. These included versions with correlated residuals between items with similar wording or meaning, consistent with recent research using the same approach [27,73], as well as models with item reduction. The final best-fitting model was subsequently tested in the clinical sample to assess model generalizability and to conduct a split-sample cross-validation.

For each model tested, the goodness-of-fit indices considered were the Comparative Fit Index (CFI), the Tucker-Lewis Index (TLI), the Root Mean Square Error of Approximation (RMSEA) with a 90% confidence interval (90% CI), and the Standardized Root Mean Square Residual (SRMR). These are additional descriptive indices to consider in parallel to the $\chi^2$-statistic to find the best model as this latter is sensitive to sample size [74,75]. Following established recommendations, models were evaluated using the following thresholds: CFI and TLI values more than or equal to 0.90 indicate an acceptable fit, and values greater than or equal to 0.95 a good fit [76], values of RMSEA below 0.06 and SRMR below 0.08 indicate a good fit and when below 0.05 an excellent fit [74,76,77].

**Reliability analyses.** Normality of score distributions for all MentS subscales and total scores was evaluated through skewness, kurtosis, and the Shapiro-Wilk test. These indices were computed separately for the clinical and control samples. Following common guidelines, distributions were considered approximately normal when skewness and kurtosis values fell within the ±2 range [78], and when the Shapiro-Wilk test was non-significant (p > .05) [79]. Shapiro-Wilk tests indicated no violation of normality for the total scores in either group. All subscales were normally distributed in the

control group, whereas in the clinical group, the Others and Self subscales significantly deviated from normality, while the Motivation subscale did not.

Internal consistency of the obtained subscales and total scale was estimated using Cronbach's α and McDonald's ω coefficients in both control and clinical samples, according to the model found to best fit our data. Cronbach's α coefficient is deemed acceptable for exploratory research when equals to or above 0.70 to 0.79, good for research or clinical purposes when equals to or higher than 0.80 to 0.89 and excellent and suitable for diagnostic purposes when equals to or above 0.90 [80,81]. McDonald's ω is considered acceptable when equals to or above 0.70, good when equals to or above 0.80 and excellent when equals to or above 0.90 [82].

Temporal stability of the scale in the control sample was evaluated through intraclass correlations coefficient between test and 1-year retest measures for total scale and subscales using average measures intraclass correlation coefficient, with a two-way mixed model and absolute agreement type.

**Convergent validity.** Total-scale scores' convergent validity was examined differently amongst samples. In the control sample, score correlations with global psychopathology severity was inspected. Whereas in the clinical sample, correlations with borderline symptomatology were used. Subscales' validity analyses were similarly examined in both samples, with correlations between MentS-Motivation and adaptative emotion regulation, MentS-Others and certainty in reflective functioning and MentS-Self and childhood trauma.

All correlations are Spearman's rho coefficients, applied consistently across both samples to ensure methodological consistency and facilitate direct comparison of correlation strengths. This choice was motivated by violations of normality observed in the clinical group and the presence of extreme values in the control group. Full correlation tables are provided in the S1 Data.

## Results

### Descriptive statistics

The control sample included 302 participants (161 females, 141 males). The clinical sample comprised 409 participants diagnosed with borderline personality disorder (BPD), attention-deficit/hyperactivity disorder (ADHD), or both. The remaining clinical sample showed substantial diagnostic heterogeneity, with BPD and/or ADHD diagnoses often accompanied by additional psychiatric comorbidities, particularly mood, anxiety, and substance use disorders. Most participants in this group were receiving at least one psychiatric medication at the time of participation. Full demographic and diagnostic information, including age statistics by gender and diagnosis, is presented in Table 1.

**Table 1. Descriptive statistics–number of participants, gender, diagnostic, age.**

| Sample | Gender | Diagnostic | N/Ntot | Age | | | |
|---|---|---|---|---|---|---|---|
| | | | | Mean | SD | Min | Max |
| **Control** | Female | N/A | 161 (/302) | 37.3 | 13.6 | 19.6 | 74.9 |
| | Male | N/A | 141 (/302) | 36.8 | 10.7 | 20.2 | 73.4 |
| **Clinical** | Female | BPD | 121 (/409) | 26.5 | 9.11 | 16 | 58 |
| | | BPD/ ADHD | 51 (/409) | 30.4 | 10.27 | 16 | 55 |
| | | ADHD | 94 (/409) | 35.7 | 12.59 | 17 | 77 |
| | Male | BPD | 12 (/409) | 27.8 | 6.93 | 20 | 42 |
| | | BPD/ ADHD | 18 (/409) | 33.1 | 9.63 | 18 | 50 |
| | | ADHD | 113 (/409) | 37.5 | 12.41 | 16 | 74 |

N/A = Not Applicable (control sample); BPD = Borderline Personality Disorder; ADHD = Attention Deficit Hyperactivity Disorder; N = Number of Participants.

## Confirmatory factor analysis

Confirmatory factor analysis (CFA) was first conducted on the control sample to evaluate the fit of the originally proposed 3-factor and 28-item model [36].

As initial model fit was not satisfactory (CFI = .868; TLI = .856; SRMR = .110; RMSEA = .116 (.110 −.121); $\chi^2$ (347) = 1745.0, p < .001), modification indices from covariance of residuals and theoretical considerations were used to improve the model. Specifically, we allowed for errors between items similar in formulation or meaning to correlate [27,73]. A revised model with 28 items and correlated residuals (3 pairs; item 3 with 5, 22, 4) showed improved fit (CFI = .923; TLI = .915; SRMR = .093; RMSEA = .092, 90% CI [.086–.097]; $\chi^2$ (343) = 1209.0, p < .001). However, inspection of the modification indices revealed that item 25 showed a large modification index of 1224.85 for its loading on the MentS-Self factor (MentS-S), and 1045.84 for the MentS-Others factor (MentS-M), indicating significant cross-loadings. These extremely high values persisted even after residual correlations were added (e.g., 1214.50 for MentS-S in model 2), suggesting that item 25 did not fit well with the underlying structure.

Given this persistent misfit and its disproportionate influence on the model, item 25 was removed in model 3. The resulting 27-item model including only 3 residual correlations, yielded the best fit in the control sample (CFI = .954; TLI = .949; SRMR = .079; RMSEA = .071, 90% CI [.065–.077]; $\chi^2$ (318) = 800.0, p < .001) and was retained.

This final model was subsequently tested in the clinical sample to assess its generalizability and conduct a split-sample cross-validation. Results also indicated an acceptable fit in the clinical data (CFI = .902; TLI = .891; SRMR = .077; RMSEA = .069, 90% CI [.064–.074]; $\chi^2$ (318) = 937.0, p < .001), supporting the replicability of the model across both samples. All analyses values are reported in Table 2.

Model 3 (27 items) showed improved fit over Model 2 (28 items), with ΔCFI = 0.031 and ΔRMSEA = 0.021, both exceeding recommended thresholds for meaningful change [83,84]. This supports the selection of the more parsimonious 27-item model.

Subsequent analyses therefore use the MentS-27 item version – omitting item 25. The scoring system is the following: Self-subscale = [8, 11, 14, 18, 19, 21, 22, 26], Others-subscale = [2, 3, 5, 6, 10, 12, 20, 23, 28], Motivation-subscale = [1, 4, 7, 9, 13, 15, 16, 17, 24, 27], The total score consists of the addition of all subscales = [MentS-S + MentS-O + MentS-M]. Items with reversed coding are: 8, 9, 11, 14, 18, 19, 21, 22, 26, 27.

## Reliability analyses

Table 3 presents the MentS mean scores, results of normality testing, and reliability analyses for the total scale and subscales in both samples.

Reliability estimates were acceptable for the Motivation-subscale in the clinical group and for the Others-subscale in the control group, and good for all other subscales and whole-scale in both samples.

**Table 2. Goodness of fit indices–confirmatory factor analyses.**

| Model | CFI | TLI | SRMR (scaled) | RMSEA (scaled) | $\chi^2$(ddl) (scaled-user) |
|---|---|---|---|---|---|
| 1) 3-factor, 28 items | 0.868 | 0.856 | 0.110 | 0.116 (.110−.121) | $\chi^2$ (347) = 1745, p < .001 |
| 2) 3-factor, 28 items, CR | 0.923 | 0.915 | 0.093 | 0.092 (0.086-0.097) | $\chi^2$ (343) = 1209, p < .001 |
| 3) 3-factor, 27 items, CR | 0.954 | 0.949 | 0.079 | 0.071 (0.065-0.077) | $\chi^2$ (318) = 800, p < .001 |
| 4) Model 3 applied to the clinical sample: 3-factor, 27 items, CR | 0.902 | 0.891 | 0.077 | 0.069 (0.064-0.074) | $\chi^2$ (318) = 937, p < .001 |

CR = Correlated Residuals; CFI = Comparative Fit Index; TLI = Tucker-Lewis Index; SRMR = Standardized Root Mean Square Residual; RMSEA = Root Mean Square Error of Approximation.

Our francophone version of the scale demonstrated good 1 year test-retest reliability with average measures intraclass correlations coefficient. The total MentS scale demonstrated excellent reliability (ICC = .877, 95% CI [.843,.904], *p* < .001). Subscale reliability coefficients were also strong: MentS–Motivation (ICC = .837, 95% CI [.792,.873], *p* < .001), MentS–Others (ICC = .806, 95% CI [.752,.848], *p* < .001), and MentS–Self (ICC = .837, 95% CI [.791,.872], *p* < .001).

## Convergent validity analyses

Correlations with neighboring constructs and MentS mean scores were examined in both control and clinical samples, respectively displayed in Tables 4 and 5 for the control and clinical samples.

## Associations in the control sample

Table 4 displays associations in the control sample. The mentalization total-scale score (MentS_Tot) shows a significant negative correlation with general psychological distress (SCL90_GSI, $\rho = -0.164$, *p* < 0,01), indicating that higher mentalization is associated with lower distress. Motivation to mentalize (MentS_M) is significantly correlated with adaptive

**Table 3. MentS scores, normality testing and reliability analyses.**

| | MentS scores | | | | | | | | | | | | |
|---|---|---|---|---|---|---|---|---|---|---|---|---|---|
| | Sample | Mean | SD | Min | Max | Skewness | Std Error | Kurtosis | Std error | Shapiro-Wilk | p | Cronbach's α | McDonald's ω |
| MentS-Tot | Control | 95.2 | 13.5 | 55 | 135 | 0.177 | 0.140 | 0.215 | 0.280 | 0.992 | 0.117 | 0.856 | 0.867 |
| | Clinical | 97.0 | 14.6 | 59 | 131 | −0.013 | 0.121 | −0.308 | 0.241 | 0.995 | 0.160 | 0.871 | 0.879 |
| MentS-M | Control | 35.6 | 6.28 | 18 | 50 | −0.263 | 0.140 | −0.263 | 0.280 | 0.990 | 0.046 | 0.789 | 0.801 |
| | Clinical | 39.2 | 6.38 | 21 | 50 | −0.411 | 0.121 | −0.326 | 0.241 | 0.977 | <.001 | 0.770 | 0.783 |
| MentS-O | Control | 33.0 | 5.27 | 18 | 45 | −0.179 | 0.140 | −0.149 | 0.280 | 0.991 | 0.070 | 0.792 | 0.798 |
| | Clinical | 35.4 | 5.82 | 17 | 45 | −0.393 | 0.121 | −0.390 | 0.241 | 0.976 | <.001 | 0.842 | 0.847 |
| MentS-S | Control | 26.6 | 6.61 | 8 | 40 | −0.168 | 0.140 | −0.265 | 0.280 | 0.990 | 0.031 | 0.824 | 0.828 |
| | Clinical | 22.4 | 6.76 | 8 | 40 | 0.301 | 0.121 | −0.352 | 0.241 | 0.987 | 0.001 | 0.808 | 0.815 |

MentS-Tot = Total MentS score; MentS-M = MentS-Motivation subscale; MentS-O = MentS-Others subscale; MentS-S = MentS-Self subscale; Mean = MentS mean scores; SD = Standard Deviation in MentS scores; Min = Minimum MentS scores; Max = Maximum MentS scores; Std Error = Standard Error; p = Shapiro-Wilk's test p-value.

**Table 4. Spearman's rho correlations in the control sample.**

| | MentS_Tot | MentS_M | MentS_O | MentS_S |
|---|---|---|---|---|
| SCL90_GSI | **−.164**\*\* | .111 | .078 | −.520\*\* |
| CERQ_Ad_ER | .282\*\* | **.183**\*\* | .184\*\* | .254\*\* |
| RFQ_c | .418\*\* | .117\* | **.218**\*\* | .602\*\* |
| CTQ_Tot | −.045 | .074 | .070 | **−.201**\*\* |
| CTQ_EmAb | .020 | .161\*\* | .161\*\* | −.221\*\* |
| CTQ_PhAb | .035 | .062 | .057 | −.052 |
| CTQ_SeAb | .042 | .124\* | .100 | −.120\* |
| CTQ_EmNeg | −.072 | .008 | .028 | −.149\*\* |
| CTQ_PhNeg | −.144\* | −.068 | −.089 | −.156\*\* |

Hypothesized correlations in bold font.

\*\*. Correlation is significant at the 0.01 level (2-tailed).

\*. Correlation is significant at the 0.05 level (2-tailed).

(CERQ_Ad_ER, $\rho=0.183$, $p<0{,}01$). Mentalizing of others (MentS_O) is significantly correlated with reflective functioning certainty ($\rho=0.218$, $p<0{,}01$). Mentalizing the self (MentS_S) correlates negatively with various trauma measures, such as emotional abuse (CTQ_EmAb, $\rho=-0.221$, $p<0{,}01$), emotional neglect (CTQ_EmNeg, $\rho=-0.149$, $p<0{,}01$), and physical neglect (CTQ_PhNeg, $\rho=-0.156$, $p<0{,}01$). The full association table can be found in the Supporting Information File as S1 Table in S1 Data.

**Associations in the clinical sample.** Table 5 displays associations in the clinical sample. Mentalization total-scale score (MentS_Tot) is significantly negatively correlated with borderline symptomatology (BSL_23, $\rho=-0.145$, $p<0{,}01$), suggesting that higher mentalization capacity is associated with fewer borderline symptoms. Motivation to mentalization (MentS_M), is significantly correlated with adaptive emotion regulation (CERQ_Ad_ER, $\rho=0.208$, $p<0{,}01$). Mentalization of others (MentS_O) shows positive correlations with reflective functioning certainty (RFQ_c, $\rho=0.266$, $p<0{,}01$). Mentalization of the self (MentS_S) reveals negative correlations with several trauma measures, including emotional abuse (CTQ_EmAb, $\rho=-0.183$, $p<0{,}01$), emotional neglect (CTQ_EmNeg, $\rho=-0.206$, $p<0{,}01$), physical neglect (CTQ_PhNeg, $\rho=-0.206$, $p<0{,}01$), and childhood trauma (CTQ_Tot, $\rho=-0.199$, $p<0{,}01$). The full association table can be found in the Supporting Information File as S2 Table in S1 Data.

## Discussion

The main purpose of this study was to test the validity of the MentS self-report questionnaire amongst francophone control and clinical participants. The large sample size in this study provides robust and well-powered psychometric results. To the best of our knowledge, it is the only the second study to include a significant sample of individuals with borderline personality disorder, making the findings comparable to first development and validation study of the MentS [36].

The three-factor structure of the MentS scale has been consistently replicated in prior validation studies [41–44,46,48,49]. Among these studies, three recommended removals of items [45–47]. In our study, the decision to recommend excluding item 25 "I can easily describe what I feel" was empirically driven from confirmatory factor analyses results in both samples. Therefore, we hypothesized this item is not interpreted consistently by participants. This observation is consistent with findings from the original MentS study [36], which indicated that item 25 loads differently in the principal components across samples in the scale's initial validation study: loading more with the "mentalizing others" dimension in the control sample whilst loading more on the "self-mentalizing" dimension in the clinical sample.

Regarding the discriminative power of the scale, our descriptive findings suggest that the low scores on the MentS-Self subscale in the clinical sample may most significantly differentiate from the non-clinical sample. This observation aligns

**Table 5. Spearman's rho Correlations in the clinical sample.**

|  | MentS_Tot | MentS_M | MentS_O | Ment_S |
|---|---|---|---|---|
| BSL_23 | **−.145**\*\* | .005 | .085 | −.425\*\* |
| CERQ_Ad_ER | .265\*\* | **.208**\*\* | .112\* | .294\*\* |
| RFQ_c | .431\*\* | .237\*\* | **.266**\*\* | .521\*\* |
| CTQ_Tot | −.026 | .060 | .082 | **−.199**\*\* |
| CTQ_EmAb | .033 | .122\* | .144\*\* | −.183\*\* |
| CTQ_PhAb | −.039 | −.018 | .046 | −.128\*\* |
| CTQ_SexAb | .066 | .110\* | .053 | −.036 |
| CTQ_EmNeg | −.099\* | −.014 | .000 | −.206\*\* |
| CTQ_PhNeg | −.145\*\* | −.092 | −.013 | −.206\*\* |

Hypothesized correlations in bold font.

\*\*. Correlation is significant at the 0.01 level (2-tailed).

\*. Correlation is significant at the 0.05 level (2-tailed).

with Dimitrijević, Hanak [36], who reported lower self-mentalizing scores in individuals with borderline personality disorder (BPD) compared to controls.

Reliability analyses yielded good overall internal consistency. Total-scale and the MentS-Self subscale can be used in clinical as well as research purposes in clinical and non-clinical populations. However, caution is warranted for the MentS-Others and especially for the MentS-Motivation subscales in clinical purposes in francophone populations. This last subscale was notably found to be below the acceptable threshold in the clinical sample in the original MentS study [36] and in the clinical and community samples in the Chinese validation study [49]. This underscores the importance of further research into the measurement of the motivation to mentalize.

The validity analyses yielded results that align with our hypotheses. First, higher levels of psychopathology in the control sample and elevated borderline symptomatology in the clinical sample were both associated with lower overall mentalization scores. This finding supports the well-documented relationship between impaired mentalization and psychopathological expression, particularly in borderline personality disorder [85]. Second, a positive association was observed between motivation to mentalize and adaptive emotion regulation, underscoring the role of curiosity and motivation as facilitators of emotional processing and regulation. Indeed, the curiosity-driven stance central to the motivation to mentalize fosters effective emotion regulation [86,87]. Third, a positive relationship between mentalization of others and reflective functioning certainty was identified, consistent with the idea that maintaining a balance between certainty and uncertainty about mental states is critical for effective mentalizing [1]. Finally, the validity analyses revealed significant negative associations between self-mentalizing and childhood trauma in both samples. This finding is in line with a recent meta-analysis which reports significant, negative associations between childhood maltreatment and mentalizing capacities [88]. Both the present validity analyses and the meta-analytic findings support the conceptualization of mentalization as a factor protecting from the development of psychopathology [89]. In this vein, self-mentalizing has been found to moderate the link between psychopathological manifestations and self-functioning [53,90]. Self-mentalizing was also found to mediate the relationship between childhood adversity and psychosis [91]. Additionally, self-mentalizing was found to be negatively and significantly associated to narcissistic vulnerability [55]. In this context, further investigation into the protective role of self-mentalizing appears warranted.

Furthermore, there is a growing recognition of the self-other distinction as critical to accurate mental state attribution [92], for which the MentS is particularly well suited to investigate given its dimensional structure which distinguishes self and other mentalizing. Research highlights the clinical relevance of contrasting dimension of mentalizing, particularly in conditions such as borderline personality disorder (BPD) [93]. More recently, the self-other distinction was established to be of transdiagnostic relevance [94]. Our current findings in clinical populations with BPD and/or ADHD tend to align with these observations. Above diagnostic conditions, an illustration of the relevance of considering the self-other distinction acutely applies to cases of severe abuse, where victims can become hypersensitive to the perpetrator's emotions, internalizing the perpetrator's wants and needs as a means of protecting oneself in abusive and neglectful developmental environments, referred to as "identification with the aggressor", a phenomena for which a specific scale was recently developed [95]. Overall, these strands of research point to the importance of further exploring the interplay between self- and other-mentalizing as a key factor in understanding mental state attribution. This nuanced approach could provide deeper insights into the mechanisms underlying mentalization impairments and guide the refinement of therapeutic strategies targeting specific mentalization dimensions.

The findings of the present study offer preliminary insights for clinicians interested in working with the concept of mentalization. First, the scale can now be employed for francophone professionals, both with clinical and non-clinical populations. Second, although the MentS is not intended for diagnostic use, initial results suggest that it may be useful as a screening tool for reduced global mentalizing capacity, and more particularly, for evaluating self- and other-mentalizing, as well as motivation to mentalize. Our results on impairments in self-oriented mentalizing points to the clinical sensitivity of this subscale, although further work should examine both low and very high scores in relation to the expression of

clinical symptoms. Finally, no formal cut-off values have been established to date, as further research is required to define clinically meaningful clinical thresholds. However, one preliminary study proposed a total score of 100.5 as a potential benchmark for distinguishing between individuals with schizophrenia and non-clinical populations [49]. From a dimensional point of view, the scale and subscales scores may help the clinician and patient formulate strengths and difficulties in mentalizing at the outset of treatment and further along the intervention to assess potential therapeutic effects. Here, test-retest clinical intervention studies are needed to evaluation whether the Ment-S constitute a valuable scale to measure therapeutic change mechanisms.

This study should be considered keeping in mind the following limitations. First, the systematic and informed exclusion professional participants in the control sample was impossible, and constitutes a limitation of the present study. Second, the use of correlated errors in structural equation modeling may have introduced bias or inaccuracies in the model estimates, necessitating caution in interpreting these findings [96]. Third, childhood trauma data relied on retrospective self-reports, such as the CTQ, are subject to important limitations. Although the CTQ is widely used and psychometrically validated, retrospective reporting of early adversity is inherently vulnerable to various sources of bias. These include memory distortions, repression or dissociation related to traumatic content, and underreporting due to stigma, guilt, or shame. Furthermore, the study did not incorporate clinical interviews, collateral reports, or biological markers, which could have enhanced the reliability and ecological validity of trauma assessment. Future research would benefit from combining self-report tools with multimodal assessments, such as clinician-administered interviews or corroborating evidence from health or forensic records, to more accurately capture trauma exposure. Finally, the use of the RFQ has limitations as recent studies suggest it may in fact measure mentalization as a unidimensional [67], or bidimensional self- and other-mentalizing construct [97], therefore calling for further refinement. Future research should address these limitations and further explore the MentS scale's clinical applications. Specifically, self-mentalizing and the self-other distinction have emerged as critical factors in understanding mentalization impairments, particularly in the context of trauma. Future studies could investigate the discriminative clinical power of the MentS and examine the role of MentS-Self scores in various diagnostic conditions.

In conclusion, the francophone validation of the MentS demonstrates strong psychometric properties, supporting its use in both clinical and non-clinical populations. It is recommended to exclude item 25 from score interpretation when using the francophone version, as its meaning may vary across populations. By distinguishing mentalization dimensions, the scale facilitates a nuanced understanding of mentalization impairments. Notably, impairments in self-mentalizing emerge as a key marker of psychopathological risk, particularly in trauma-affected clinical populations. This highlights the critical role of the self-other distinction, where reduced self-mentalizing, coupled with heightened focus on others, may blur identity boundaries and negatively affect the embodied sense of self, underscoring its relevance for clinical applications. These findings collectively validate the utility of the MentS scale for capturing key dimensions of mentalization and highlight its potential for furthering our understanding of psychopathology and therapeutic interventions.

## Supporting information

**S1 Data.** **S1 Table.** Full Table 4 Correlations in control sample - Spearman's rho Correlations. **S2 Table.** Full Table 5 Correlations in clinical sample - Spearman's rho Correlations. **S3 File.** French translation of The Mentalization Scale (MentS). **S4** List of researchers who contributed to this work as part of the RF-TBM Consortium.
(DOCX)

## Acknowledgments

The PI (Martin Debbané) was funded by the Swiss National Science Foundation (Grant No. 100014_179033). The funders had no role in study design and data collection, analyses, or interpretation.

## Author contributions

**Conceptualization:** Nader Ali Perroud, Martin Debbané.

**Data curation:** Eléonore Pham, Rosetta Nicastro, Eva Rüfenacht, Nader Ali Perroud, Martin Debbané.

**Funding acquisition:** Martin Debbané.

**Methodology:** Flora Descartes, Vincent Besch, Margaux Bouteloup.

**Supervision:** Martin Debbané.

**Writing – original draft:** Flora Descartes.

**Writing – review & editing:** Vincent Besch, Margaux Bouteloup, Nader Ali Perroud, Martin Debbané.

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
