## [Decision Letter · Decision Letter 0]

3 May 2025

PONE-D-25-06596Validation of Mentalization Scale (MENT-S) in francophone control and clinical samplesPLOS ONE

Dear Dr. Descartes,

Thank you for submitting your manuscript to PLOS ONE. After careful consideration, we feel that it has merit but does not fully meet PLOS ONE’s publication criteria as it currently stands. Therefore, we invite you to submit a revised version of the manuscript that addresses the points raised during the review process.

We look forward to receiving your revised manuscript.

Kind regards,

Marco Innamorati

Academic Editor

PLOS ONE

Journal Requirements:

5. We note you have included a table to which you do not refer in the text of your manuscript. Please ensure that you refer to Table 1 in your text; if accepted, production will need this reference to link the reader to the Table.

6. We notice that your supplementary tables 5, and 6 are included in the manuscript file. Please remove them and upload them with the file type 'Supporting Information'. Please ensure that each Supporting Information file has a legend listed in the manuscript after the references list.

Reviewers' comments:

Reviewer's Responses to Questions

**Comments to the Author**

1. Is the manuscript technically sound, and do the data support the conclusions?

Reviewer #1: Partly

Reviewer #2: Yes

Reviewer #3: Partly

2. Has the statistical analysis been performed appropriately and rigorously? 

Reviewer #1: Yes

Reviewer #2: Yes

Reviewer #3: Yes

3. Have the authors made all data underlying the findings in their manuscript fully available?

Reviewer #1: Yes

Reviewer #2: Yes

Reviewer #3: No

4. Is the manuscript presented in an intelligible fashion and written in standard English?

Reviewer #1: Yes

Reviewer #2: Yes

Reviewer #3: No

5. Review Comments to the Author

Reviewer #1: This study evaluated the psychometric properties of the French version of the Mentalization Scale (MentS) in both community and clinical samples. A total of 711 participants, including individuals with borderline personality disorder (BPD), ADHD, and co-occurring BPD and ADHD, completed the scale. Confirmatory factor analysis supported a 27-item, three-factor structure (mentalizing self, mentalizing others, and motivation to mentalize) as optimal for both groups. According to the Authors, the French version of the MentS is suitable for research and clinical use.

The topic of the paper is both timely and interesting. However, there are several important points that require attention. I recommend a thorough and very careful revision of the manuscript to address these issues and strengthen the overall quality of the work.

In the Introduction, the Authors present the RFS scale in a somewhat cursory manner. It would have been beneficial to provide a more comprehensive overview of the RFS, including discussion of related instruments such as the RFQ-8. Notably, the authors do not address several critical issues—frequently highlighted in the literature—regarding the challenges of measuring mentalization with the RFS. As a result, the transition to the MentS scale feels abrupt and insufficiently justified. Furthermore, it is important to acknowledge that, both before and after the development of the MentS, other instruments for assessing mentalization have been introduced. Including references to these alternative measures would have strengthened the context and provided a more balanced perspective.

In the Introduction (lines 80–92), the authors assert that Fonagy and Luyten have addressed and resolved the methodological limitations outlined earlier (lines 77–79). They also cite encouraging data supporting the francophone version of the RFQ. However, the subsequent transition to introducing the Mentalization Scale (MentS) feels abrupt and insufficiently justified. While the MentS is indeed a valuable tool for multidimensional mentalization assessment, the authors do not adequately articulate its comparative advantages over existing measures. A more thorough discussion is needed to clarify why the MentS was selected over alternative instruments, particularly in light of its strengths (e.g., multidimensional structure) and potential limitations. Highlighting how the MentS addresses gaps left by other tools would provide a stronger rationale for its use and enhance the coherence of this section.

In my view, the description of the MentS in the manuscript is insufficient and does not offer readers a clear understanding of the instrument. The structure and dimensions of the MentS are not adequately defined, while greater emphasis is placed on the interpretation of scores. Providing a more detailed explanation of the scale’s underlying structure and its specific dimensions would significantly enhance the reader’s comprehension and better contextualize the meaning and relevance of the reported scores.

I found the section of the Introduction addressing convergent validity (lines 128 to 134) to be somewhat confusing, possibly due to an oversimplification of the authors’ argument. The formulation of multiple hypotheses in the latter part of the introduction is particularly perplexing, as several of these hypotheses, especially the one regarding a negative correlation between the MentS-S scale and self-reported childhood trauma, are not logically substantiated within the text. Additionally, I am concerned by the decision to assess childhood trauma using a self-report measure, as this approach raises questions about the

A seemingly minor point concerns the way references are cited within the text. While I understand that the authors may have chosen this citation format to simplify the preparation of the reference list, it does not conform to any recognized citation styles typically used in scientific writing. Adhering to a standard citation style is important for clarity and consistency throughout the manuscript.

Method

In describing the sample, the authors state that they recruited participants for the control group via the Prolific website. Based on my understanding, Prolific is a platform that compensates research participants for their time and contributions. While this approach facilitates rapid and diverse recruitment, it does raise the question of whether such a method increases the likelihood of enrolling so-called “professional participants”—individuals who frequently participate in online studies for compensation. It is still possible for samples to include individuals with substantial prior experience in online research, which may influence their responses or introduce certain biases. While Prolific offers valuable tools to mitigate the risk of recruiting predominantly “professional participants,” researchers should remain mindful of this potential limitation and consider reporting the level of participant experience in their sample description.

Including a brief description of the MentS and reporting Cronbach's alpha values at the end of the Participants paragraph is unusual and disrupts the logical flow of the manuscript. The Participants section should focus on describing the sample's characteristics and recruitment methods, as well as any relevant demographic information. Details about the instruments used—including their structure, dimensions, and psychometric properties such as reliability (e.g., Cronbach's alpha)—are more appropriately placed in the Materials or Measures section.

In summary, the inclusion of the MentS description and its Cronbach's alpha values in the Participants paragraph is misplaced. This information would be more appropriately presented in the section dedicated to describing the study's materials or measures, where readers expect to find details about the instruments and their psychometric properties.

Additional measures

The section describing additional measures administered alongside the MentS lacks clarity regarding the rationale for selecting these instruments. It is important for the authors to explain why each measure was included and how it relates to the study’s objectives or hypotheses. Without this context, readers may find it difficult to understand the relevance and contribution of these additional assessments.

Furthermore, the information about the translation into French, which appears at the end of this paragraph, seems out of place. Details about translation procedures are typically presented in a dedicated section on instrument adaptation or within the description of the specific measure being translated. Placing this information in the "Additional Measures" section disrupts the logical flow and organization of the manuscript.

To improve clarity and coherence, the authors should clearly justify the inclusion of each additional measure and relocate the translation details to a more appropriate section of the manuscript.

As a minor point, it should be noted that the authors used the RFQ-8, the short form of the Reflective Functioning Questionnaire, rather than the longer version of the instrument.

Statistical analysis

Based on the results of the confirmatory factor analysis (CFA) conducted on both the control and clinical samples, the authors decided to eliminate item 25 from their version of the MentS. After removing item 25, the authors used this abbreviated version of the MentS for subsequent reliability analyses. Then they computed the associations between the measures of interest in both samples. The lack of clarity regarding the MentS scoring system in the manuscript creates significant ambiguity, particularly in interpreting the meaning of positive or negative values.

The fact that the authors used the Jamovi software for statistical analyses should have been stated at the beginning of the paragraph, not at the end of the section dedicated to the CFA.

Discussion

In this section of the manuscript, the authors' extensive focus on findings derived from the Childhood Trauma Questionnaire (CTQ) short form—a 28-item self-report measure—warrants further scrutiny. Although the CTQ is a validated instrument, its brevity and reliance on retrospective self-reporting raise concerns about its ability to comprehensively assess complex and multifaceted experiences of childhood maltreatment. The CTQ evaluates five subscales (physical, emotional, and sexual abuse, along with emotional and physical neglect), yet such a condensed format may lack the depth needed to capture the nuances of traumatic experiences, including contextual factors, chronicity, and subjective impact. Given these limitations, the heavy emphasis on CTQ-based results in the discussion may inadvertently oversimplify the interpretation of childhood trauma's role in the study's outcomes.

It is worth to note that self-report measures are vulnerable to recall bias, social desirability bias, and underreporting, especially for stigmatized experiences like abuse. Participants may consciously or unconsciously minimize or deny traumatic events. The CTQ quantifies maltreatment severity but provides no qualitative insights into the lived experience of trauma. Critical factors such as developmental timing, relational dynamics, and coping mechanisms remain unaddressed.

Finally, the CTQ’s subscales (physical/emotional/sexual abuse, physical/emotional neglect) show high intercorrelations, making it difficult to isolate specific trauma types. This overlap complicates interpretations of how distinct maltreatment experiences relate to outcomes like mentalization deficits.

The brief mention of the study’s limitations in the final section of the discussion is inadequate. A thorough and transparent discussion of limitations is essential for contextualizing the findings, acknowledging potential biases, and guiding future research. Simply referencing these issues without elaboration does not provide readers with a clear understanding of how the study’s design, measures, or sample characteristics may have influenced the results. Expanding this section to address specific methodological constraints, such as the reliance on self-report measures, sample representativeness, or the generalizability of the findings, would strengthen the manuscript and enhance its scientific rigor.

Reviewer #2: Although the authors justify model refinements (e.g., residual correlations and removal of item 25), a brief expanded discussion on clinical interpretation and usability of the shortened 27-item scale would benefit readers.

Implications for practitioners using this tool in diagnostic or therapeutic settings could be elaborated—especially given the growing interest in mentalization in clinical psychology.

A few language polishing points (minor grammar or flow) may help improve readability, though these are not major.

Reviewer #3: The present study explores the psychometric properties of the francophone translation of the MentS, performing a confirmatory factor analysis and evaluating test-retest reliability.

The study is sound. However, I would like to draw your attention to some areas for improvement.

First of all, the acronym MENT-S appears in the title, while MentS is used throughout the manuscript. I recommend choosing one version and using it consistently.

In general, I suggest reviewing spelling according to a consistent language style (British or American English, e.g., behavior / behaviour).

I also recommend that the authors revise the writing and the English language throughout the manuscript, as some sentences are difficult to understand or seem incomplete, and there are a few mistakes. Below are some examples:

- Line 22 ("it's" should be "its"?)

- Lines 33–34 (revise sentence structure)

- Line 48 ("may more likely to report increased")

- Line 58 ("The centrality of mentalizing in human" → "mentalization")

- Lines 150–151 (combine the two sentences)

- Lines 161–162 (combine the two sentences)

- Line 165 ("sample composed of" → "sample was composed of")

- Line 252 ("analysis were" → "analysis was")

- Lines 308–309 (split the sentence using a full stop → ". Results ...")

- Line 352 (some values are reported as r=xxx, others as r = xxx. Decide on spacing and ensure consistency)

I also recommend avoiding the use of asterisks in the text to indicate p-values (it is acceptable in tables, but in the text it is always preferable to report values fully, e.g., p < 0.001).

Furthermore, in table- section "note", when acronyms are explained, the initial letters of the words should be capitalized (e.g., ADHD = Attention Deficit Hyperactivity Disorder; RMSEA = Root Mean Square Error of Approximation).

Abstract: I suggest reporting the test-retest reliability values in parentheses and naming at least the constructs that were assessed when referring to "additional measures" (line 40).

Methods:

- Lines 173–175 would be more appropriate in the Results section.

- I would move the following line (inclusion criteria at the end of line 169).

- Before describing the various measurement instruments, insert a subheading titled Measures.

- The section on missing data should be placed under Data Analysis and should specify whether any missing data were present. Also, provide a reference for the rule “If less than 30% of item values were missing”.

- Line 291–292: specify why the non-parametric Spearman coefficient was used (were the variables not normally distributed?). Additionally, there is a lack of detail regarding outliers, multicollinearity, heteroscedasticity, etc.

While the general approach for statistical analyses is sound and report a comprehensive set of fit indices was reported, several statistical concerns and methodological considerations should be addressed to enhance the rigor and interpretability of the results.

- The use of ML estimation may not be optimal given that questionnaire data are typically ordinal in nature. ML assumes multivariate normality and continuous data, which is often violated in Likert-type scales. A more appropriate estimation method would be Weighted Least Squares Mean and Variance adjusted (WLSMV), which is robust for ordinal variables and commonly recommended in such contexts.

- The final model incorporates 24 correlated residuals, which is a substantial number relative to the total number of items (28). While the justification provided (i.e., similar item wording) is acknowledged, such an extensive modification raises concerns about overfitting and may artificially inflate model fit. Correlated errors should be added sparingly and only when strong theoretical justification is present. I read that you have brought this observation within limitation of the study.

- After introducing model modifications and removing item 25, it would be important to statistically compare the models (e.g., using Chi-square difference tests or ΔCFI/ΔRMSEA) to support the decision to retain the revised structure. Additionally, cross-validation using split samples or bootstrapping could help assess the stability of the modified model.

- you briefly mention the use of Spearman’s rho due to non-normal distributions but, as previously mentioned, other essential assumptions such as outliers, multicollinearity, and heteroscedasticity are not addressed. Clarification on these points would strengthen the statistical validity of the analyses.

Given the large number of residual correlations (24 pairs) needed to improve model fit in the confirmatory factor analysis, I wonder whether you considered using an Exploratory Structural Equation Modeling (ESEM) approach. ESEM could have allowed for more flexibility in modeling item cross-loadings without relying on post-hoc correlated error terms, and might have provided a better representation of the underlying structure. Including a rationale for not choosing this approach, or a brief discussion of its potential relevance, would strengthen the methodological justification of the CFA strategy adopted.

6. PLOS authors have the option to publish the peer review history of their article (what does this mean?). If published, this will include your full peer review and any attached files.

Reviewer #1: No

Reviewer #2: No

Reviewer #3: No

---

## [Author Response · Author response to Decision Letter 1]

17 Jul 2025

PONE-D-25-06596

Validation of Mentalization Scale (Ment-S) in francophone control and clinical samples

PLOS ONE

Rebuttal letter – Answer to the comments

Each of these items with mentioned labels have been uploaded.

Journal Requirements:

We have looked at the journal’s style requirements and have done modifications consequently to meet the journal’s style requirement for file naming.

We hereby confirm that our data cannot be publicly shared based on the study’s ethics committee (Project-ID 2021-00694), but we do make it available upon request for a specific collaboration, which will entail a special permission granted by the local ethics committee.

There are ethical and legal restrictions preventing the sharing of a de-identified dataset, as such sharing is not permitted under the ethics agreement approved by the cantonal ethics committee (Project-ID 2021-00694). Although authorization was granted to collect and analyze the data, sharing the dataset is prohibited due to the sensitive nature of the personal information involved, linked to the rules governing data collection locally in the hospital setting. The ethics committee prohibits the sharing of data. As a result, the dataset cannot be made publicly available, only upon request for a specific collaboration, which will entail a special permission granted by the local ethics committee.

Address of the cantonal authority on human medical research:

Commission cantonale d'éthique de la recherche (CCER)

Rue Adrien-Lachenal 8

1207 Genève

Phone: +41 22 546 51 01

E-mail: ccer@etat.ge.ch

There are restrictions as stated above. Therefore, we did not update the Data Availability statement. As it already stated in the original submission: All relevant data are within the manuscript and its Supporting Information files.

There are ethical and legal restrictions on sharing a de-identified data set, as sharing data is prohibited under the current ethics agreement approved by the cantonal ethics committee. While authorization was granted to collect and analyze the data, sharing the dataset is not allowed due to the sensitive nature of the personal information involved. Consequently, as stated above, the dataset must remain confidential and cannot be shared.

Address of the cantonal authority on human medical research:

Commission cantonale d'éthique de la recherche (CCER)

Rue Adrien-Lachenal 8

1207 Genève

Phone: +41 22 546 51 01

E-mail: ccer@etat.ge.ch

We have amended our list of authors to ensure that it reflects the institution where the work was done. Additionally, we have added Dr Eva Rüfenacht from Geneva University Hospitals who was also part of the data collection as well as the RFTBM consortium (Réseau Francophone de Thérapie Basée sur la Mentalisation) in the Supporting Information file. This addition exhaustively and more precisely reflects involvement in the work done and data collected.

5. We note you have included a table to which you do not refer in the text of your manuscript. Please ensure that you refer to Table 1 in your text; if accepted, production will need this reference to link the reader to the Table.

We thank you for your attention to this error and have made modifications accordingly.

6. We notice that your supplementary tables 5, and 6 are included in the manuscript file. Please remove them and upload them with the file type 'Supporting Information'. Please ensure that each Supporting Information file has a legend listed in the manuscript after the references list.

We thank you for your attention to this error and have made modifications accordingly.

Reviewers' comments:

Reviewer's Responses to Questions

Comments to the Author

1. Is the manuscript technically sound, and do the data support the conclusions?

Reviewer #1: Partly

Reviewer #2: Yes

Reviewer #3: Partly

2. Has the statistical analysis been performed appropriately and rigorously?

Reviewer #1: Yes

Reviewer #2: Yes

Reviewer #3: Yes

3. Have the authors made all data underlying the findings in their manuscript fully available?

Reviewer #1: Yes

Reviewer #2: Yes

Reviewer #3: No

4. Is the manuscript presented in an intelligible fashion and written in standard English?

Reviewer #1: Yes

Reviewer #2: Yes

Reviewer #3: No

5. Review Comments to the Author

Reviewer #1:

• This study evaluated the psychometric properties of the French version of the Mentalization Scale (MentS) in both community and clinical samples. A total of 711 participants, including individuals with borderline personality disorder (BPD), ADHD, and co-occurring BPD and ADHD, completed the scale. Confirmatory factor analysis supported a 27-item, three-factor structure (mentalizing self, mentalizing others, and motivation to mentalize) as optimal for both groups. According to the Authors, the French version of the MentS is suitable for research and clinical use.

The topic of the paper is both timely and interesting. However, there are several important points that require attention. I recommend a thorough and very careful revision of the manuscript to address these issues and strengthen the overall quality of the work.

In the Introduction, the Authors present the RFS scale in a somewhat cursory manner. It would have been beneficial to provide a more comprehensive overview of the RFS, including discussion of related instruments such as the RFQ-8. Notably, the authors do not address several critical issues—frequently highlighted in the literature—regarding the challenges of measuring mentalization with the RFS. As a result, the transition to the MentS scale feels abrupt and insufficiently justified. Furthermore, it is important to acknowledge that, both before and after the development of the MentS, other instruments for assessing mentalization have been introduced. Including references to these alternative measures would have strengthened the context and provided a more balanced perspective.

In the Introduction (Lines 80–92), the authors assert that Fonagy and Luyten have addressed and resolved the methodological limitations outlined earlier (Lines 77–79). They also cite encouraging data supporting the francophone version of the RFQ. However, the subsequent transition to introducing the Mentalization Scale (MentS) feels abrupt and insufficiently justified. While the MentS is indeed a valuable tool for multidimensional mentalization assessment, the authors do not adequately articulate its comparative advantages over existing measures. A more thorough discussion is needed to clarify why the MentS was selected over alternative instruments, particularly in light of its strengths (e.g., multidimensional structure) and potential limitations. Highlighting how the MentS addresses gaps left by other tools would provide a stronger rationale for its use and enhance the coherence of this section.

We thank the reviewer for this helpful and constructive comment.

The line numbers mentioned in our responses refer to the clean version of the manuscript.

In the revised manuscript, we expanded the paragraph originally spanning lines 78 to 90 to provide a more comprehensive and fine-grained presentation of the Reflective Functioning Scale (RFS), also mentioning the well-documented limitations in assessing mentalization (lines 74 to 82).

This section is now complemented by a similarly concise yet thorough overview of the Reflective Functioning Questionnaire (RFQ), focusing specifically on the 8-item version (lines 90 to 97)

We then briefly introduce earlier self-report measures developed prior to the MentS, namely the Mentalization Questionnaire (2012) and the Mentalized Affectivity Scale (MAS) (2017), as well as the mention of other more recent self-report developed tools since the MentS (lines 97 to 108).

Finally, at the end of this revised section, once all relevant tools have been introduced and the MentS is appropriately situated among them, we inserted a new paragraph assessing elements specific to the MentS in comparison to existing instruments (lines 109 to 113).

• In my view, the description of the MentS in the manuscript is insufficient and does not offer readers a clear understanding of the instrument. The structure and dimensions of the MentS are not adequately defined, while greater emphasis is placed on the interpretation of scores. Providing a more detailed explanation of the scale’s underlying structure and its specific dimensions would significantly enhance the reader’s comprehension and better contextualize the meaning and relevance of the reported scores.

In the revised version of the manuscript, we have added elements to provide a better understanding of the MentS instrument. Specifically, we now include a clear explanation of each dimensions composing the structure of the MentS, as well as example items for each subscale. We further make explicit the number of items per subscale (lines 113 to 129) and provide the full French-translated instrument as supplementary information to the article.

• I found the section of the Introduction addressing convergent validity (lines 128 to 134) to be somewhat confusing, possibly due to an oversimplification of the authors’ argument. The formulation of multiple hypotheses in the latter part of the introduction is particularly perplexing, as several of these hypotheses, especially the one regarding a negative correlation between the MentS-S scale and self-reported childhood trauma, are not logically substantiated within the text.

We thank the reviewer for this constructive observation. We acknowledge that the section on convergent validity may have appeared confusing due to an overly condensed presentation of our rationale.

In the revised manuscript we have clarified and expanded this s

---

## [Decision Letter · Decision Letter 1]

3 Sep 2025

Validation of the Mentalization Scale (Ment-S) in francophone control and clinical samples.

PONE-D-25-06596R1

Dear Dr. Descartes,

We’re pleased to inform you that your manuscript has been judged scientifically suitable for publication and will be formally accepted for publication once it meets all outstanding technical requirements.

Kind regards,

Marco Innamorati

Academic Editor

PLOS ONE

Additional Editor Comments (optional):

Reviewer #1:

Reviewer #3:

Reviewers' comments:

Reviewer's Responses to Questions

**Comments to the Author**

1. If the authors have adequately addressed your comments raised in a previous round of review and you feel that this manuscript is now acceptable for publication, you may indicate that here to bypass the “Comments to the Author” section, enter your conflict of interest statement in the “Confidential to Editor” section, and submit your "Accept" recommendation.

Reviewer #1: All comments have been addressed

Reviewer #3: All comments have been addressed

2. Is the manuscript technically sound, and do the data support the conclusions?

Reviewer #1: (No Response)

Reviewer #3: Yes

3. Has the statistical analysis been performed appropriately and rigorously? 

Reviewer #1: (No Response)

Reviewer #3: Yes

4. Have the authors made all data underlying the findings in their manuscript fully available?

Reviewer #1: (No Response)

Reviewer #3: No

5. Is the manuscript presented in an intelligible fashion and written in standard English?

Reviewer #1: (No Response)

Reviewer #3: Yes

6. Review Comments to the Author

Reviewer #1: (No Response)

Reviewer #3: (No Response)

7. PLOS authors have the option to publish the peer review history of their article (what does this mean?). If published, this will include your full peer review and any attached files.

Reviewer #1: No

Reviewer #3: No

---

## [Editor Report · Acceptance letter]

PONE-D-25-06596R1

PLOS ONE

Dear Dr. Descartes,

I'm pleased to inform you that your manuscript has been deemed suitable for publication in PLOS ONE. Congratulations! Your manuscript is now being handed over to our production team.

Kind regards,

on behalf of

Dr. Marco Innamorati

Academic Editor

PLOS ONE